# Towards Pragmatist Thermodynamics: An Essay on the Natural Philosophy of Entropy and Sustainability

**DOI:** 10.3390/e27121257

**Published:** 2025-12-15

**Authors:** Carsten Herrmann-Pillath

**Affiliations:** Max Weber Centre for Advanced Cultural and Social Studies, Erfurt University, 99089 Erfurt, Germany; carsten.herrmann-pillath@uni-erfurt.de

**Keywords:** lived thermodynamics, Peirce, entropy production, out-of-equilibrium assemblages, Lotka’s maximum power, sustainability

## Abstract

Classical thermodynamics (CT) has become integrated into everyday life, especially through its applications in engineering. In contrast, out-of-equilibrium thermodynamics (OET) is often viewed as a fundamental science that seems distant from daily experiences. While “energy” is a familiar term in households, “entropy,” which refers to degraded energy, remains enigmatic. This gap in understanding has significant implications for developing effective sustainability practices. CT typically emphasizes the efficiency of individual systems that produce work, often overlooking the entropy production that occurs within larger, interconnected systems. This paper aims to establish a philosophical framework that transforms OET into what is referred to as “lived thermodynamics.” This framework is grounded in pragmatism, particularly drawing from the early synthesis of thermodynamics and evolutionary theory proposed by Charles S. Peirce. A central aspect of this approach involves shifting the focus from traditional “systems” to out-of-equilibrium assemblages. In these assemblages, the physical trends of entropy production are often interrupted and redirected by evolutionary innovations and random events. The evolving envelope of open systems within these assemblages manifests an increasing rate of entropy production. This synthesis of thermodynamics and evolutionary theory builds on Lotka’s pioneering contributions and contemporary theories, particularly Vermeij’s work on the evolution of power. The framework introduces a sustainability criterion based on entropy. By applying this criterion, OET can evolve into “lived thermodynamics,” fostering a holistic understanding of energy use in devices and technological systems while considering the broader implications of entropy production in the out-of-equilibrium assemblages in which we live.

## 1. Introduction

Thermodynamics emerged as a scientific discipline that examines physical phenomena permeating our daily experiences, much like gravity. However, there is a significant disparity in how thermodynamics is received by the public [1]. While the concept of energy is well known and has gained prominence due to recent technological advancements, such as electric cars, the idea of entropy remains largely unfamiliar or is often seen as mysterious. Even physics teachers often fail to connect thermodynamics with daily life in their didactics [2,3]. In reality, entropy just represents degraded energy, a concept people encounter when they feel the heat produced by their electronic devices. This disparity in public reception of closely related physical concepts is even more pronounced when comparing the understanding of classical thermodynamics (CT) with recent developments in out-of-equilibrium thermodynamics (OET). The former ties up with people’s everyday lives because it is used in the engineering of familiar technologies, such as combustion engines. In contrast, OET is almost entirely unknown, even though it underlies the very essence of life as we know it and the natural systems that constitute our surroundings. Indeed, in his classic work on entropy, Percy Bridgman asserted that thermodynamics is a discipline with a “smell of human origin” [4]. This was written before the ascent of OET.

This asymmetry is essential for understanding the current ecological crisis and the responses of human societies. A key observation is that CT emphasizes the importance of efficiency in relating the concept of entropy to everyday human life. In simple terms, this means minimizing waste. However, within the context of OET, this viewpoint fails to consider the complex, non-linear feedback mechanisms that govern efficiency-driven actions in systems composed of multiple subsystems and hierarchical levels. As a result, backfire, also known as the rebound effect, becomes highly probable [5,6]. In general, CT primarily suggests a first-law perspective in practical applications, while OET emphasises the second law, moving beyond merely treating entropy as a “waste-producing” physical process. OET highlights the role of energy dissipation and entropy production in driving the formation of structures, a concept first introduced by Prigogine in his theory of dissipative structures [7]. This perspective is crucial for analytically balancing the physically determined phenomena of growth in out-of-equilibrium open systems with the related processes of entropy production.

The paper goes beyond these observations by presenting a pragmatist approach to thermodynamics. The gap between advances in thermodynamics in the sciences and its application in everyday life is not merely a matter of scientific education, such as teaching physics in high schools [8]. What is lacking is a systematic conceptual framework that bridges the gap between theory and practice. This gap reflects the separation between the sciences and the social sciences in theorizing thermodynamics, despite many attempts to synthesize these perspectives [9]. The social sciences, particularly economics, can create a theoretical connection between physics and human life if they are combined with thermodynamics. However, due to the fragmented nature of the social sciences, we require a comprehensive philosophical framework to unite them. I refer to this framework as “pragmatism,” using the term in two distinct senses.

The first aspect highlights the historical connections between CT and the science of the steam engine. It examines how we can perceive current sustainability challenges as similar to notable instances of what can be termed “Lived thermodynamics.” This concept refers to the systematic relationship between the everyday experience of thermodynamic processes and the scientific advancement of thermodynamic theory. The gap between lived experience and theory is narrower for CT than for OET, which is why I focus on the latter in this paper.

In her comprehensive history-of-science account of the rise of thermodynamics, Cara Daggett paints a picture we interpret as “Lived thermodynamics” [10]. In Victorian England, this new physical theory was intertwined with the ethics of work and with Protestant beliefs that saw waste and idleness as sinful. Many Scottish scientists were Presbyterians, and the emerging concept of energy took on almost theological implications, unlike Darwinian theory, which often faced religious skepticism. Energy was closely associated with work and was regarded as beneficial for humanity, driving the forces behind the industrial transformation of the world. In contrast, entropy was often linked to Christian apocalyptic imagery, suggesting an ultimate unravelling of all human efforts. However, science provided solutions by advocating for relentless efforts to minimize entropy—defined as wastefulness and idleness—through the efficient organization of labor in capitalist factories. This new understanding of energy included optimizing human work and its reproduction. In other words, thermodynamics became a crucial component of what Max Weber described as the process of capitalist rationalization. The alternative would have been to leave dissipation to nature, which would contradict the biblical notion of man’s superiority, often viewed through a gendered lens. In this context, the mastery of nature is portrayed as both tragic and integral to the human experience.

As we observe, CT had not only emerged as a scientific discipline but also became part of everyday experience in Victorian England, or “Lived thermodynamics.” This paper raises the question of whether and how the modern thermodynamics of out-of-equilibrium open systems can become “lived” in a manner similar to CT in the 19th century. I argue that we are still constrained by the 19th-century framework, despite many scientific fields advocating for the relevance of new approaches for decades. This situation is partly influenced by significant contributions that have popularized thermodynamics, while also explicitly excluding OET from the discussion [11]. On the other hand, there are contributions that move in the opposite direction and even incorporate aspects of quantum theory [12]. Both extremes highlight that the gap between OET and human experience remains unbridged. To a large extent, this gap reflects a broader disconnect between science and everyday life, which is highlighted by the significant divide between the formal language of mathematics and the natural languages we use daily—a point also made by Percy Bridgman. In the 19th century, the term “efficiency” became part of common vernacular, whereas today “entropy” remains closely associated with increasingly complex mathematical theories in the sciences.

A notable example of building lived thermodynamics of OET can be found in Wessling’s book [1]. This work intertwines biographical elements with reflections on current crises, while introducing the fundamental concepts of OET. Wessling engages in a continuous dialogue between theory and practical experience, illustrating how OET can help tackle today’s ecological challenges. The book also emphasizes the dangers of making major policy mistakes by overlooking the foundational principles of OET in real-world applications. It proposes a new understanding of sustainability, which I will explore further in this paper.

This example leads to the second sense of “pragmatism.” It refers to pragmatism as a philosophical discipline that presents specific epistemological, ontological, and metaphysical principles and highlights the critical role of practical experience in advancing knowledge [13]. Pragmatism was developed by American philosophers in the 19th and early 20th centuries, including influential figures such as William James and John Dewey. Notably, thermodynamics is relevant to the work of only one of these founding figures, Charles Sanders Peirce. Peirce is particularly significant for our argument because he established connections between thermodynamics and Darwinian theory [14,15]. This relationship was also acknowledged by Ludwig Boltzmann, although it was only systematically explored later by Alfred Lotka [16,17]. Despite his contributions, Peirce is rarely referenced in contemporary discussions regarding this connection [18]. I aim to address this gap by arguing that Peirce’s philosophy provides a fundamental ontological perspective on thermodynamics. This perspective can bridge the distinction between the life-world and scientific theory, suggesting that action serves as the foundation of scientific knowledge. Knowledge must be actionable and demonstrated through actions that tackle the challenges humans face. The ontological linkage to recent advances in thermodynamics is established by considering action as physical work and relating work to a thermodynamic understanding of agency [19,20,21,22]. However, in this paper, I do not pursue this foundational line of thinking further.

In contemporary society, one of the significant challenges we face is the eco-social transformation aimed at achieving sustainable living practices. This transformation requires establishing a new relationship between the human domain and nature and overcoming the misplaced duality that reflects cultural misconceptions about human autonomy and mastery over nature. The integration of the theoretical principles of understanding both domains, often referred to as the ontological continuity thesis [23,24], can be achieved with a synthesis of thermodynamics and evolutionary theory. Important steps in this direction have already been made, with magistral works such as Vaclav Smil’s [25] on growth or Vermeij’s [26] view on power as the driver of evolution, and the contributions of many leading ecological economists, such as Daly [27] and Gowdy [25]. Economics holds significant relevance, particularly because neoclassical economics originated from 19th-century scientific thought and evolved into a framework for understanding the “living economy” in the 20th century. CT profoundly influenced Paul Samuelson’s landmark mathematical modelling of market equilibrium theory [26]. This shows that mainstream economic views implicitly incorporate thermodynamics, even though this physical perspective on real-world economic processes has been largely overlooked. Therefore, we can explore how a similar formal analogy might enhance the material application of OET in fostering eco-social transformation.

This paper begins with a brief overview of the three pivotal ideas in the Peircean synthesis, randomness, evolution, and finality. The discussion builds on my extensive monographic treatment of these issues [27]. Randomness is the non-probabilistic concept of propensities in the Popperian sense, evolution is a creative process that results in growing constraints on random change (dubbed “habits”), from which forces of finality emerge. Section 3 introduces a new view on OET which is informed by assemblage theory which has been so far mainly employed in the social sciences. Pragmatist thermodynamics eschews the “systems” view that has defined the ontology of thermodynamics thus far and relates OET to assemblages of open systems, thereby grounding its integration with evolutionary theory. The Second Law, as recently reformulated by Lineweaver in this journal [28], imposes finality on such assemblages, which can be seen as envelopes of successive open systems. I show that the assemblage approach dovetails with recent efforts of developing a “mesoscopic non-equilibrium thermodynamics” which aims at explaining dynamic processes of self-assembly (in this journal [29]). Section 4 introduces the critical conceptual link between theory and lived thermodynamics, which is Alfred Lotka’s principle of maximum power. In terms of related scientific disciplines, lived thermodynamics is most salient in economics, where mainstream economics is more closely aligned with CT, and OET associates with ecological economics. Section 5 presents an important application: a thermodynamic criterion for sustainability. This criterion focuses on minimizing the acceleration of the rate of entropy production on Earth that cannot be exported through radiation. In practical terms, this aligns with Wessling’s criterion, which involves selecting the option that produces the least entropy among available alternatives or discarding those options that would contribute to increased entropy during their generation. Section 6 concludes by sketching some consequences of PT for our societies.

## 2. Peirce and the Origins of Pragmatist Thermodynamics

Pragmatism is a philosophical approach that systematically links knowledge, particularly scientific knowledge, to action. When considering the integration of thermodynamics with human life, it becomes clear that the development of scientific knowledge is fundamentally rooted in experience. A prominent example of this is the role of the steam engine in advancing new theories during the 19th century. Furthermore, this perspective emphasizes the systematic importance of experiments in generating knowledge, which aligns well with scientific methodologies like Popperian fallibilism [30]. The unity of science and experience is rooted in the shared goal of solving real-life problems. We interpret this as a normative commitment to make science intelligible and applicable in the real world. As we argued in the introduction, CT certainly qualifies as a paradigmatic case of this normative view in science.

In this section, I briefly summarize the most relevant variant of philosophical pragmatism, which has its origins in Peirce’s philosophy [31], and which prepares the ground for what I call “pragmatist thermodynamics” (PT). Peirce drew systematic conclusions from the most recent scientific developments of his time, specifically thermodynamics and evolutionary theory. We highlight three key ideas: randomness as a metaphysical principle (“tychism”), evolution, and final causality.

### 2.1. Randomness and Evolution

Peirce’s view on tychism presents a metaphysical perspective in which the world is characterized by the random generation of novelty (which he calls “chance”) (without referring to Peirce, this is close to “ontic openness” in [32]). This idea aligns with the Darwinian concept of variation in the logic of variation, selection and retention. His perspective contrasts sharply with many modern approaches to randomness, particularly those in information–theoretic frameworks that focus on Shannon entropy and are based on probability. Peirce’s understanding of randomness is similar to Popper’s view of propensity in an open universe [33]. Propensities cannot be reduced to numerical probabilities; rather, the latter can be interpreted only as subjective probability assignments. Peirce regards statistics as a key tool of scientific inquiry by induction; however, he rejects subjectivist notions of chance. Probabilistic induction serves as a practical way to navigate a radically open and uncertain world, a strategy commonly employed in economics. It also plays an important role in the recent controversies about the Maximum entropy approach in OET, where the followers of Jaynes [34] emphasize its exclusive role as a method of predicting the evolution of out-of-equilibrium systems [35] (often referred to as MaxEnt, which is also used to refer to a statistical method [36]), whereas others interpret the formalism as relating to material thermodynamics [37,38]. If we deploy a Peircean notion of propensity in eschewing intrinsic probabilistic notions of randomness, this conceptual tension can be resolved [27].

It is important that Peirce’s view on randomness is independent of modern quantum theory, which is often invoked in contemporary discussions on entropy. Quantum theory presents a significant rupture between theoretical concepts and the everyday world, making it problematic from the perspective of PT. By receiving Peirce and the notion of propensity, we can set aside the quantum perspective, although, of course, this does not imply that we consider it scientifically invalid. In the context of evolutionary theory, it is sufficient to acknowledge the simple fact that there is an immense space of possible events, in which individual events are nearly impossible, or have close to zero probability [39]. This vast space emerges from the exponentially growing number of variations that can recombine existing variants into new assemblages, which in turn define further trajectories of possible events [40] (for more detail on these arguments, see [41]). This indicates that, in evolution, the state space itself is continually evolving. This source of irreducible randomness, or Peircean chance, is conceptually independent from quantum physics.

In the realization of propensities, time is essential, particularly synchronicity. This reconciles Peirce’s tychism with efficient causality. All events have a cause (which differs from “causeless” events in the quantum realm), but this does not mean we can predict what will happen. The key lies in the unpredictability of co-occurrences—previously independent causal trajectories that converge at a certain point in time and thereby lead to the emergence of new recombinations. The hypercomplexity of the manifold of causal trajectories in the world renders the single event truly random in the sense of algorithmic incompressibility [42] and the thermodynamic depth of the causal chains [43].

For Peirce, randomness reflects the creativity of the world. Events occur spontaneously, resulting in novel assemblages of elements. Peirce [44] (p. 221) explicitly opposes chance to probabilistic interpretations of the Second Law, which imply that the flow of events tends towards the distribution of highest probability: In contrast, chance fixes a state of the world, thus balancing the forces of randomization. We can relate this process to the modern concept of path-dependence since the newly emerging assemblages create constraints on further evolutionary change. In modern evolutionary theory, Peirce’s view comes close to Gould’s [45] alternative to the new synthesis in Darwinism. At a conceptual level, the most probable state towards which energy dissipation leads is increasingly limited by various constraints. This limitation on evolutionary trajectories imposes finality.

### 2.2. Finality

Peirce defends the concept of final causality at this point. Final causality is a highly contentious term in modern science and is mostly rejected. However, Peirce presents a methodologically sound version of finality that builds on the distinction between particulars and universals or types [14]. In biology, we cannot predict which variant will emerge from mutations, but we can exclude mere variational possibilities [46]. While we cannot predict what kind of person a newborn will become, we can predict with certainty that the baby will not have wings and will not be able to fly after four years of development. Evolution accumulates a growing and increasingly complex set of constraints that channel evolutionary trajectories in a certain direction, which constitutes finality [47]. However, this does not mean that the evolutionary trajectory simply repeats itself over time. For example, mutations can open up new trajectories, implying that other sets of constraints may become relevant and redirect the process.

Peirce defines final causality in terms of types that are realized by a variety of possible trajectories leading toward this end. This means he combines stochasticity and finality. This idea aligns straightforwardly with thermodynamics: the gas in an isolated container will achieve an equiprobable distribution of molecules, but we cannot predict the locations of individual molecules based on causal laws. Finality is established by the Second Law of Thermodynamics, which determines a specific outcome from a vast array of diverse instances. Schrödinger made a significant error by assuming that complex life is improbable. He overlooked the fact that the forms of life that actually exist have evolved under increasingly complex constraints, making these realized states the most likely outcomes [48]. I believe this reasoning highlights the limitations of probabilistic approaches to understanding evolution. Referring back to Peirce’s argument, the Second Law of Thermodynamics affects the final outcome of an equiprobable distribution. However, in the context of evolution, what truly matters is the chance occurrence of a single molecule arriving at a specific location, as this can significantly influence subsequent events.

This distinction aligns with Vermeij’s critique of probabilistic reasoning within the theory of evolution [49]. Vermeij argues that, in evolution, averages do not drive processes; instead, it is the realized states and entities that represent directional trends of increasing power through selection. These entities impose mutual constraints on one another. This will be explored further in Section 4. For example, the emergence of top predators initiates synergistic processes in ecosystem evolution, ultimately leading to greater ecosystem productivity.

In PT, finality is established through the acceleration of entropy production along a trajectory of endogenously evolving constraints. Peirce refers to these constraints as “habits,” which is a central term in pragmatism [50]. A habit is a recurrent and regular pattern that channels the flow of events in the world, including natural laws, in Peirce’s view. Natural laws can be understood as the most general form of a constraint that defines the boundary between what is possible and what is impossible. A key question arises: how is a habit preserved and reproduced? Evolutionary theory provides an answer through the concept of stasis [45]. Despite the fact that evolutionary processes are open and indeterministic, most of these processes result in the reproduction of constraints via final causality, as described above (for a detailed exposition of this view, see [47]).

In summary, PT utilizes a Peircean metaphysics and ontology as the foundation for applying theoretical principles of OET to real-world phenomena, with the goal of addressing the challenges that humans face. The pillars of this foundation are a non-probabilistic notion of chance (“tychism”), the evolution of regular patterns in terms of habits, and finality as directedness of evolutionary change. This philosophical basis is crucial because a significant methodological challenge in deploying OET lies in reconciling the established ontology of thermodynamics with the diverse real-world structures that differ from its typical applications. For example, an ecosystem cannot be directly compared to a liquid being stirred, as the patterns of mixing in a liquid—driven by energy dissipation—are distinct, as illustrated in Prigogine’s theory. The ontological structures differ significantly; plant organisms are the components of an ecosystem, while molecules constitute a liquid. By adopting Peirce’s framework, we can methodologically justify these cross-domain applications of OET.

## 3. The Foundational Pragmatist Move: From Systems to Assemblages

### 3.1. The Many Meanings of Entropy

The difference between CT and OET reflects the fundamental distinction between isolated, closed, and open systems. CT pertains to isolated systems and conceptualizes entropy as a characteristic of their equilibrium state. Consequently, many physicists argue that applying the concept of entropy outside this context is meaningless. However, in recent decades, there has been a significant increase in the various uses and interpretations of the term “entropy” across different disciplines within the natural sciences and even extending into the human and social sciences. This has led to considerable confusion regarding even the most basic concepts [32]. These extensions can be classified into three categories, which may overlap:The first concept is the extension of OET, which focuses on the processes of entropy production beyond isolated systems. A key reference in this area is the work of Prigogine and his colleagues on dissipative structures in closed systems [51]. Dissipative structures are still viewed as single systems that reach a certain end state defined by minimum entropy production. However, more recently, various approaches to open and non-linear systems have been proposed, often using the term “maximum entropy production,” which seems to create tension with earlier perspectives [52].The second significant development is the merging of Shannon’s information theory with thermodynamics, for which Boltzmann’s definition of entropy laid the groundwork [53]. This integration has been a crucial driver in the development of a wide variety of mathematical measures of entropy, also incorporating applications of entropy in quantum mechanics. All these approaches are grounded in the fundamental concept of probability.The third aspect involves extending concepts beyond physical systems to include chemical and biological systems, particularly in the ecological sciences [7]. These extensions can be divided into two main focuses: we can emphasize mathematical forms of entropy, such as measures of biodiversity [36], or concentrate on the physical processes associated with entropy production [54].

From the perspective of PT, these recent developments have created a significant gap between scientific advancements and the human experience. The various uses and meanings of entropy are often elusive and challenging to relate to everyday human actions. We argue that this gap can be bridged by drawing appropriate conclusions from the shift from CT to OET. A key distinction lies between information-theoretic approaches and those that focus on entropy production, since in case of the latter, this distinction is often blurred and creates controversies [35,55]. Following many critics in the literature, PT emphasizes the materiality of entropy and the continuity between CT and OET, placing less importance on information-theoretic perspectives unless they explicitly address materiality, such as Landauer’s principle in the energetics of computation [56]. This continuity enables a seamless integration of lived experiences surrounding entropic processes.

### 3.2. The Concept of Out-of-Equilibrium Assemblages

The transition from classical thermodynamics CT to OET is crucial for understanding the interconnectedness of life, as humans exist not in isolated systems, but in complex networks of open systems operating at various scales. While planet Earth can be viewed as a closed system with minimal exchange of matter with outer space, this fundamental reality is often not adequately reflected in traditional thermodynamic theories. These theories typically impose the concept of a “system” onto a specific entity, whether it’s a stirred liquid in a glass, a living cell, or the entire planet.

This use of the term “system” is a critical issue. Most applications of thermodynamics focus on a single “system,” yet this reference can vary widely in scale. For instance, a steam engine is analyzed through the lens of CT, while an ecosystem is assessed using OET. However, when considering the steam engine’s use by humans, we must recognize it exists within a broader context, which can also be viewed as a system, such as the human economy. This larger “system” should be treated as an open system, comprising a multitude of diverse systems that are not fully aligned or congruent with the economy, including the surrounding ecological systems.

There are two key implications for this perspective. First, by considering the thermodynamics of the encompassing system, we can investigate the roles of the constituent systems. Second, due to the absence of systemic congruence, we need to analyze the relationships between these various systems, which may at times be internal to a specific system and at other times external. For example, a farm can be conceptualized as a thermodynamic system in which parts of the larger embedding ecosystem are internalized via the design of the farmer [57], yet other parts have an impact on the farm processes externally, such as migratory birds who might play havoc with the harvest [58].

From this brief discussion, we draw an important conclusion: we should avoid using the term “system” in the context of OET and instead refer to OET as the study of *out-of-equilibrium assemblages*. When relating OET to the reality humans experience, there is no universally applicable method for navigating the complexity of assemblages and identifying a “system.” This challenge is central to the empirical validation of OET theorems, such as those based on the Maximum Entropy Production Principle [59]. However, this is not a setback; rather, it paves the way for integrating OET with human experience. The concept of sustainability can only be meaningfully applied to an organic farm if we consider not only its internal production processes but also its integration within larger ecosystem networks [1]. This perspective is crucial when assessing its impact on biodiversity. However, once we take this step, we can no longer unequivocally identify a single larger system in which the organic farm is embedded.

The concept of “assemblage” provides a unifying framework for thermodynamic theories and subjects related to human societies [60,61]. We follow De Landa’s approach here as it builds on a thorough realist ontology that is compatible with the sciences. An assemblage is an arrangement of heterogeneous elements that maintain an internal relationship with each other (“interiority”), fostering the emergence of causal powers at the level of the assemblage while remaining autonomous in relation to the environment of the assemblage (“exteriority”). Thus, unlike a system, the external relationships are not fully determined by internal mechanisms, which control the boundary between the interior and the external environment (see Figure 1).

As we will explore, a crucial consequence of this perspective is that we cannot expect theoretical hypotheses of OET to apply unequivocally to the real world, since they mathematically refer to a system where interiority and exteriority are clearly and unequivocally separated to render the formalism analytically solvable. Instead, they should be understood as trends that rarely fully materialize, as they are constantly disrupted, interrupted, and redirected throughout the evolution of assemblages and the impacts of Peircean chance. Hence, we cannot simply assume that assemblages show a clear tendency in entropy production; we must take into account the constraints and finalities that result from their interactions, which never add up to a single and coherent system.

The concept of assemblage is essential for linking OET to the human lifeworld. However, is this concept also relevant to OET research itself? Is it simply a passing fad introduced by a philosopher? Would adopting this concept ultimately imply a departure from systems-based formalism? A particularly interesting example of OET research that clearly demonstrates the value of the assemblage view is the emerging mesoscopic research, which empirically refers to particles, molecules, and similar entities that actively drive the formation of material structures in a given environment [63]. In this “mesoscopic nonequilibrium thermodynamics” MNET the key term is “self-assembly.” Instead of using the term “assemblage,” the phrase “non-equilibrium self-assembling NESA structures” is proposed, which I recommend as a definitional specification of “assemblage” in the context of MNET. Hence, this is not just a terminological resemblance; it highlights that there is a starting situation in a certain process, such as gelation, where we cannot speak of a “system” in terms of fixed boundaries maintained by internal structures comprising both the particle and a segment of the environment. In social science, assemblage theory employs the Deleuzean term “body without organs,” which refers to a seed entity from which assemblage formation begins. Similarly, in MNET, the concept of Janus particles is essential as they are the drivers and seeds of self-assembly. The term encompasses a wide variety of particles with different properties, such as magnetic, optical, or amphiphilic, which generate distinct patterns of interaction with the environment [64].

The concept of Janus particles establishes a primordial concept of agency that emerges from distinct thermodynamic processes shaping these interactions (Deacon [47] refers to these phenomena as “morphogenesis”). In the chemical context, Janus particles exhibit autonomous mobility, meaning that, in a larger environment, the tendency towards equilibrium cannot fully manifest. This leads to the significant conclusion that this environment does not achieve systemic closure. The active properties of Janus particles arise from their potential for energy dissipation and entropy production by exploiting the chemical energy of the environment, as seen in catalysis [29]. Through surface gradients, this autonomous potential induces their movements and fluxes in the environment, maintaining the larger assemblage out of equilibrium.

The concept of self-assembly involves heterarchic interactions between particles and their environment. First, the movements of the particles change the structure of the environment, and second, this structure co-evolves with the particles’ movements. This structure can be described using a set of transport coefficients, such as fluidity or viscosity [65]. An important corollary is that we cannot rely solely on generic notions of entropy production that refer only to a systemic or macroscopic level. Principles such as maximum entropy production can be seen as forms of structurally conditioned finality in the Peircean ontology. However, this does not imply that extremum principles are realized or that equilibria are achieved, as long as energy sources are not consumed, which reflects the state of nature outside of human artifacts such as laboratories or engines. MNET researchers have proposed several specifications for NESA structures, including the integrative concept of an “effective potential.” This concept replaces the general principle of minimizing free energy potential. The main distinction is the inclusion of entropy production derivatives that depend on the structural parameters of the evolving NESA structure.

The main point of this discussion is that we can draw a direct conceptual analogy between MNET research and social science assemblage theory, similar to the previously mentioned analogy between CT and neoclassical economics equilibrium theory. Like other approaches that relate OET to higher ontological levels, such as ecosystems or human economies, the challenge remains in how to directly apply a mathematical formalism like that developed in MNET research to these levels.

There is another strand of thermodynamics research that formalizes the analogical approach, particularly in simulation methods related to the maximum entropy production principle. This is primarily applied in ecological systems research, where ecological networks can be modeled through computer simulations [66]. The empirical validity of these models is established by comparing simulation results with real-world patterns. However, these models still require establishing boundaries within the reference framework, such as simulating a “pond.” At the same time, they allow the assembly of network connections through the actions of components such as bacteria or phytoplankton. An important observation here is that we can now distinguish between three different levels of language in the PT approach to assemblages: the formal mathematical structures of thermodynamics, the modeling algorithms used in simulation methods, and the natural language of social sciences.

In summary, PT proposes a theoretical shift from the systems view to the assemblage view when connecting theory to the realities of human life, referred to as “Lived thermodynamics.” The assemblage view is particularly relevant for applying OET to real-world cases where fixed boundaries of systems cannot be determined. Therefore, a critical methodological issue in PT is how to relate the three levels of language that we have now identified to one another.

### 3.3. The Three Languages of PT

An important insight for my argument is that conceptual analogies bridge the gap between a conventional systems-based approach to thermodynamics and the real-world structures in which humans live. This is in addition to the artificial contexts where this approach results in pragmatically meaningful and useful insights and applications. We can even approach both the terms “system” and “assemblage” in a productive way.

Two key questions emphasize important aspects of developing a PT perspective on OET. First, we can ask about the function of a component system within a larger system: What are steam engines used for? Second, we should consider the relationship between the larger system that utilizes steam engines and other non-congruent assemblages, such as the ecological system in which the economy operates. In the case of the steam engine, one might argue that its purpose is determined by human intentions. However, this assumption suggests that humans can design and manage the entire set of interacting and non-congruent systems across all scales and scopes of assemblages, which is clearly not the case. Therefore, examining steam engines from a PT perspective on OET requires us to investigate their functions and interdependencies beyond human design, within the broader context of embedding and interacting out-of-equilibrium assemblages (Figure 2).

There is a fundamental difference between CT and OET in how they conceptualize a “system” when transitioning from theory to the real world. Open systems are characterized by more than just fixed boundaries that separate their internal mechanisms from their external environment. In practice, since OET specifically refers to open systems, it can only be meaningfully applied to assemblages of open systems. Conventional thermodynamics typically distinguishes between the system and its environment. However, in OET, the environment itself comprises other open systems arranged in complex assemblages. These assemblages are open, but not causally closed, meaning they are not solely determined by internal causal mechanisms. Instead, they operate under external constraints, with the ultimate influences defined at the planetary level. Specifically, open systems create constraints for other open systems.

As a result, PT emerges as an integrative framework in two senses. First, PT integrates CT and OET via assemblage theory, and second, assemblage theory integrates theory and Lived thermodynamics.

Figure 3 outlines the conceptual framework of integrative PT. We differentiate between two levels: the theoretical level and the practical level, which we refer to as “Lived thermodynamics.” The central concept in PT is “assemblage.” These two levels are linked through formal correspondences between assemblages, as theorized in PT, and assemblages as the constituent ontological units in the real world.

The three levels of the diagram relate to different types of language. The theoretical level employs formal mathematical language to construct logically coherent theories, which implies a mathematical notion of “system.” PT is divided into two main strands: CT and OET, with “equilibrium” serving as the reference point in both, directly connecting to the formal notion of a system.

In contrast, Lived thermodynamics fundamentally differs from theoretical thermodynamics in that it uses natural language, making PT practically relevant and epistemically accessible to people. There are two central bridging concepts that have widespread repercussions in Lived thermodynamics. The bridging concept in CT is “efficiency,” which is universally applied in fields such as economics and engineering. The bridging concept in OET is “entropy production,” which, so far, has been marginal in lived thermodynamics.

The formal correspondence between the theoretical level and lived thermodynamics is established by employing various methods of modelling and simulating assemblage dynamics, such as agent-based modelling. In Lived thermodynamics, we distinguish between three types of constituents that refer to the two strands of PT. First, there are real systems with fixed boundaries analyzed through CT, which primarily consist of human artifacts such as machines. In this case, the bridging concept of efficiency also establishes the formal correspondence between the two meanings of “system”: as a formal mathematical framework and as a real-world system. Second, there are natural systems that cannot be described and analyzed in terms of systems, but rather as assemblages. We specify these as natural assemblages, where living actors are the constituent units. For example, a market functions as a natural assemblage in which humans are living actors. In the real world shaped by human technology, assemblages of both artificial systems and natural assemblages play a central role, such as a river ecosystem affected by significant human engineering and economic functionality.

In summary, PT is a complex meta-theoretical framework that encompasses various thermodynamic theories as well as different ways of representing reality through both formal and natural languages. This framework is connected through conceptual analogies, such as bridging concepts. In the next step, we will elaborate on the fundamental Peircean synthesis of evolution and thermodynamics in terms of modern theory to complete the framework.

## 4. Evolution and Entropy Production in Pragmatist Thermodynamics

### 4.1. Evolution and the Second Law

In recent developments in thermodynamics, the systematic relationship between thermodynamics and the theory of evolution has been at least partly recognized. Alfred Lotka is generally regarded as the intellectual father of that synthesis [17,67]. Evolutionary theory does not consider single open systems but rather a multitude of systems that compete for access to resources, with energy serving as the ultimate and most universal unifying concept of resources. We will elaborate on this in Section 4.2. At this point, we want to highlight the fundamental transformation of thermodynamic theory that accompanies this synthesis. Evolutionary theory addresses energetic flows within assemblages of living systems and offers a new perspective on the role of entropy in these processes. This shift moves from viewing entropy as a state variable in equilibrium to focusing on entropy production in assemblages that exhibit continuously changing trajectories of change, following the Second Law in the overarching envelope of these [68]. These dynamics are driven by evolutionary processes of variation, selection, and retention.

This transformation was recently outlined by Lineweaver, who presented a version of the Second Law as it applies to OET [28]: It states that entropy production accelerates. The standard version of the law typically only allows for comparisons between two states of a trajectory: the initial state and the end state. It posits that the end states always have at least as much, and often more, entropy than the initial state. However, in evolving assemblages of open systems, end states are rarely realized; instead, they are continuously interrupted and redirected. Therefore, we need a conceptual frame that describes the resulting envelope of evolutionary change in thermodynamic terms. This perspective allows us to reconcile the divergence between the minimum entropy production principle and the maximum entropy production principle. In the context of organisms, where we observe the phenomenon of senescence, we can conceptualize this as a process of ageing that leads to the minimization of entropy production [69]. Evolution occurs through sequences of reproducing organisms, where competition and selection lead to increased entropy production. If we define the Second Law of Thermodynamics in terms of the acceleration of entropy production, time becomes significant in relation to synchrony and coincidence. Organisms competing against each other typically reproduce during periods of accelerating entropy production, expending energy in their struggle for survival. The outcome of differential reproductive success results from singular events in the life cycles of individual organisms.

Prigogine’s analysis of dissipative structures adopts a systems perspective, suggesting that a system ultimately reaches a state of minimum entropy production. In his framework, entropy production follows a trajectory where the development of dissipative structures initially leads to increasing entropy production, which only stabilizes at the final stage. When we examine collections of coevolving open systems, these processes are consistently interrupted by the causal effects of external trajectories competing for overlapping energetic resources that contribute to entropy production. This implies that when we consider the overall landscape of coevolving dissipative open systems, the evolutionary dynamics create a ratchet effect. As a result, most assemblages tend to evolve with increasing entropy production [70]. A necessary condition for this process is the continuous introduction of novelty in the composition of the assemblages. This idea aligns with Lineweaver’s reformulation of the Second Law for evolutionary assemblages: evolution represents a trajectory of increasing entropy production, effectively leveraging the Second Law as it is formulated for isolated systems. If entropy production increases, time becomes a critical factor in evolutionary change. This is because the accidental synchrony of interactions among specific open systems matters, as discussed in the context of path-dependent dynamics (see Figure 4).

These considerations imply a radical reversal of the causal analysis in the transition from CT to OET. In CT, the increase in entropy is the result of the Second Law of Thermodynamics. In OET, however, entropy production is the driving force behind the evolutionary dynamics that lead towards the end state envisioned by CT. Nevertheless, in closed systems like planet Earth, this end state is never fully realized because there is a continuous inflow of energy that sustains entropy production within various open systems on the planet. In other words, the influx of solar energy drives the evolution of sequences of open systems—organisms—which, as a totality of assemblages, accelerate the workings of the Second Law on the planet. In Lineweaver’s [28] catchwords, this shift of causal analysis is like from “We-Eat-Food” to “Food-Has-Produced-Us-to-Eat-It”.

Taking the steam engine as an example again, CT views the single steam engine as a paradigm. In contrast, OET examines the steam engine as a component within a broader assemblage of open systems, such as the British economy in the mid-19th century. By adopting an evolutionary perspective, we see these assemblages as evolving through variation, selection, and retention. In this framework, steam engines co-evolve with their surrounding assemblages, undergoing continuous technological change in terms of both design and application. At this point, we can recognize the connection between Lineweaver’s reformulation of the Second Law and the human experience. A key insight was provided by Jevons in his analysis of the coal question [72,73]. Jevons argued that improvements in steam engine efficiency, from a CT perspective, would not lead to absolute savings in coal usage; rather, the opposite would occur. Efficiency gains would result in an increased consumption of coal due to the widespread adoption of steam engines across the British economy, driven by competitive market forces. Consequently, the coal-powered economy would accelerate energy dissipation and entropy production.

While Jevons was not correct in his direct assessment of the conclusions regarding the British economy—since the evolutionary dynamic ultimately moved the economy away from steam engine technology—if we consider the broader context of evolutionary disruptions, we can see how this leads to the current crisis of the fossil fuel-based economy, in which coal remains a significant driver of entropy production, even in the case of the posterchild of renewables, China.

In conclusion, if we combine assemblage theory and the theory of evolution, we can extend Lineweaver’s reformulation of the Second Law to cases in which systemic closure cannot be reasonably assumed as a valid framework for interpreting empirical observations. These cases form the majority of natural assemblages in which human artifacts are embedded, as outlined in Figure 3. In the next move of our argument, we further enrich the PT approach to evolution to strengthen link between theory and Lived thermodynamics.

### 4.2. The Centrality of Maximum Power

In recent decades, numerous contributions have connected evolutionary theory with thermodynamic concepts, suggesting a trend across different types of systems toward an increasing energy throughput and dissipation. In addition to the maximum entropy literature we mentioned earlier, here are some notable examples, with no claim on completeness:Chaisson’s theory [74,75] suggests that there is an increasing intensity of free energy rate per unit mass, a concept that has also been embraced in big history research [76]. The essence of this analysis is that evolutionary trends can be observed even on a cosmological level, which allows for the categorization of distinct entities—such as organisms and human technological devices—on the scale of increasing energy flows through their unit mass. Additionally, there are variations of this analysis that focus on the evolution of human technology, emphasizing the increasing flow of energy through spatial units (power density) [77].This research connects to studies that examine evolution within the context of transformations of Planet Earth. These studies reveal specific trends in energy transitions toward greater complexity, increased intensity of dissipation, and innovations in utilizing the potential of inflowing solar energy for growth [71,78]. These perspectives include visions of transforming the planet into a “hybrid” structure [79], in which technological evolution would eventually shift the planet from a closed to an open system, specifically by enabling the export of material waste to outer space [80].There are various theories that claim differences among themselves, but they generally share basic principles and propose adding a new thermodynamic law to the established four laws. For example, the “Constructal Law” suggests that flow systems evolve in a direction that maximizes flow throughput by minimizing obstacles to those flows, and has been employed on the evolution of energy systems [81,82].On the topic of biological growth, there is a long-standing genealogy of theories suggesting systematic trends toward increasing size and complexity of organisms [83]. This idea is often summarized in the hypothesis that evolution favors the emergence of organisms and their related social units, which compete for the enhanced production of power in the sense of physical work, and embodied capacities to generate work [49,84].Finally, there are contributions that link these approaches to the evolution of human technology. This connection is already present in “big history” research, but it is more directly related to evolutionary theory, particularly in the field of evolutionary economics, which builds on the legacy of Georgescu-Roegen [25,85,86].

All these theories trace back to the initial formulation of an additional thermodynamic law for living systems and evolution, although they do not always make this explicit. This is associated with Alfred Lotka’s maximum power principle MPP, as it was later dubbed by Odum in his seminal synthesis of ecology and social sciences [87,88]. In our brief overview of PT, we focus on the core of Lotka’s argument which basically matches the previous argument as depicted in Figure 4. It begins by examining the effects of selection among competing organisms. Their metabolism can be understood thermodynamically as the process of harvesting energy resources from the environment to perform work. This work is directed towards their own survival and reproduction. The fundamental argument is that when these organisms compete for limited resources, those that can maximize energy flows and accumulate energy will tend to have greater reproductive success. Consequently, organisms evolve to create structures that represent low entropy states, storing free energy while simultaneously dissipating this energy through their metabolism and the pressures of evolutionary competition.

This analysis suggests that when we consider the envelope of evolutionary trajectories in out-of-equilibrium assemblages, there are two fundamental, though correlated, phenomena: growth and the increasing rate of entropy production. It is important to recognize, as mentioned previously, that we should avoid the “system fallacy” here. Growth is a phenomenon distributed across various sizes of organisms, the scope of their adaptive interactions, and hierarchical scales [89]. This means that Lotka’s theorem applies to assemblages and cannot be exactly observable in a clearly defined system, both spatially and temporally. Therefore, we talk about trends rather than final states, which are rarely achieved in practice.

I notice that Lotka’s concept of evolution, while informed by biology, is abstract and can be applied to various evolutionary processes, particularly in technology, if interpreted through the lens of evolutionary dynamics [90]. Therefore, we can propose an ontological continuity between the evolution of the biosphere and the technosphere, based on fundamental physical processes of entropy production [78,91]. This continuity also serves as a foundation for Lived thermodynamics, as it transcends the duality between nature and culture, which often implies that human life is separate from nature due to technological mastery. At the same time, approaching the technosphere as an evolutionary phenomenon helps us to avoid the illusion of control often maintained by humans who see themselves as designers and creators of technology.

Lotka’s MPP can be generalized in the context of physical theory, establishing a direct connection between thermodynamics and accumulation of biological information [21]. An organism constantly faces environmental challenges that it must meet and resolve. Kauffman [19] introduces the notion of an autonomous agent that can generate recurrent thermodynamic work cycles to maintain itself. In the context of evolution, we consider sequences of autonomous agents that work for differential reproduction. The notion of an autonomous agent complements the evolutionary logic of variation, selection, and retention by focusing on the goal of maximizing power in competition with other agents. Autonomous agents work, and thereby create constraints on other agents, which are the stuff from which the build-up of structures is pushed forward [47]. In principle, this view expands the concept of active particles that we have discussed previously in the context of MNET, thus resulting in a consistent extension of thermodynamic reasoning across different levels of physical and biological processes.

Vermeij’s [49] recent contributions on the evolution of power support this reasoning. If we define power as the embodied capacity to generate work, the evolutionary tendency to maximize power manifests across all levels and mechanisms of evolution. This framework can be applied to organisms or artefacts and can encompass both biological and cultural evolution. The concept of evolution is no longer limited to its biological context in phylogeny; it is especially relevant to the human economy. In fact, we can even reverse this argument: biological evolution can be interpreted through an economic lens [92]. Accordingly, as Vermeij argues, we can ground PT on two closely related, but different, concepts: One is the logic of variation, selection and retention, and the other is the notion of agency. The two concepts are independent because the evolution of embodied agency goes along with what has been called the “De-Darwinization” of evolution in the sense of autonomous internalization of external selective forces [93]. Agents become autonomous forces of evolutionary change.

At this stage, Vermeij presents a crucial argument for understanding the relationship between propensities and finality. A fundamental truth about biological and economic evolution is the intrinsic relationship between evolution at lower and higher levels of organization. Following Lotka’s insights, selection at a lower level encourages the emergence of mechanisms that maximize power. Within the context of CT, these processes also generate entropy as a necessary byproduct. In biological systems, this encompasses the waste produced by metabolism. While power-maximizing organisms produce more waste, a stable environment often drives a tendency toward efficiency, corresponding to the minimum entropy production principle [94]. However, power maximization does not necessarily mean that selection favors the most efficient organisms, as the dimension of time comes into play. Random events that pose significant survival risks often require the mobilization of power that exceeds optimal efficiency.

As a result, evolution creates a trend toward maximizing entropy production alongside the trend of power maximization, as we have argued in the previous section. What occurs at higher levels? Vermeij notes that evolution also unfolds through discovering new methods for exploiting waste as a resource. Various theoretical models capture this interdependency, such as Ulanowicz’s [95] analysis of ascendency, which builds on the chemical hypercycle of autocatalysis, or the economic concept of positive externalities [23]. If newly emerging methods for utilizing entropy production productively support power-maximizing entities, the entire process enhances productivity at higher levels—in the simplest terms, ecosystem productivity, as opposed to organism-level productivity. This dynamic results in a positive, non-linear feedback across levels, which further accelerates entropy production and creates distinct forms of finality.

In sum, Lotka’s MPP is the essential theoretical bridge between thermodynamic theory and Lived thermodynamics since it applies for evolution in general. Evolutionary processes unfold on all levels of physical and biological organization, and beginning with elementary forms of life, this includes agents capable of purposeful action leveraging entropy production via the pursuit of power in competition with other agents. Overall, this process has led to increased entropy production on Earth within the limits of the highest level of entropy production, specifically the radiation balance of the closed system. However, the rapid pace at which the human economy has augmented entropy production while generating waste that cannot be recycled in the necessary time frame for evolutionary adaptations in the biosphere has resulted in a severe ecological crisis [96].

## 5. Sustainability and Entropy

As I explained in Section 1, failing to adopt the OET perspective results in a misplaced emphasis on efficiency and energy in the Lived thermodynamics of humans, leading to a misguided focus on lower-level processes while overlooking the consequences at higher levels in terms of entropy production. This is particularly problematic when considering the rapid evolutionary dynamics of the human economy. Evolutionary economists have long argued that market competition can be analysed in terms of the general theoretical principles of variation, selection, and retention [97,98,99]. When we combine this with Lotka’s principle, we can gain a better understanding of the strong tendencies toward accelerated energy throughput from the onset of the Industrial Revolution. These tendencies continue to grow, with the most recent example being the rapid evolution of Artificial Intelligence (AI).

This situation is intriguing because AI itself is modelled on an evolutionary paradigm, which can be understood as a statistical process of information extraction and retention. Following Landauer’s discovery of the relationship between energy and units of information, we have learned that the construction and operation of software on hardware are governed by the laws of thermodynamics [56]. Currently, there is widespread recognition of the growing need to supply AI with more energy. This has led to a transition toward a practical understanding of thermodynamics. However, this transition is mostly limited to the design and operation of computers and data centers that consume rapidly increasing energy levels. In fact, we face a Jevons Paradox: as engineering efficiency improves, demand for AI services continues to grow. Numerous dilemmas arise at higher levels of this issue [100,101]. For instance, supplying AI for various human technologies will require a significant surge in renewable energy investments, which could further intensify the conflict between human resource consumption and the ecological health of our planet. In summary, the technological transition to AI as a general technology cannot be sustainable at the higher assemblage level, as it accelerates entropy production within the limits of the closed Earth system. This reality is often overlooked when considering solar energy as a solution to enhance the productivity of the Earth system, particularly given its higher efficiency compared to photosynthesis, which creates more leeway for growth [102]. Ultimately, this expansion must also lead to increased material entropy production, particularly through the destruction of biodiversity on Earth due to intense competition for space [103].

What can PT contribute to solving these challenges to sustainability? There are numerous definitions of sustainability. In general, sustainability refers to production processes that also maintain their production capacities over time, thereby not depleting their resources. For instance, the fossil fuel economy is not sustainable because it relies on a resource that cannot be replenished through the production processes it enables. However, from the perspective of the previous analysis, industrialization can be viewed as an evolutionary trajectory that removes constraints on the dissipation of the energy stored in fossil fuels, thereby accelerating energy dissipation [104].

Lotka’s theorem poses a dilemma when applied to a closed system like planet Earth. In this planetary context, evolution is driven by the continuous inflow of solar energy, theoretically allowing for unlimited growth in work output. However, because the system is closed, evolution faces constraints related to the construction and maintenance of the material structures that generate work, as well as the sequences of autonomous agents involved. The acceleration of entropy production leads to the accumulation of material waste in the planetary environment, most notably in the form of CO_2_ emissions, which contribute to the greenhouse effect and increase heat generation.

In other words, we can envision a further acceleration of entropy production by enhancing the technological means to harness solar energy. Indeed, photovoltaics are much more efficient and productive than photosynthesis in transforming solar energy into work. However, this comes at the cost of increasingly accumulating material waste. It is important to note that this waste cannot simply be neutralized by the reduced CO_2_ emissions. There is a high probability that the total material costs associated with expanding solar energy production by technological means will not resolve the dilemma of unlimited growth on a planetary scale.

Such interdependencies have been highlighted in holistic measures of energy use across entire energy systems, like the supply chains of photovoltaics and their embodied energy, which is specified as EROI, or energy return on investment [105]. However, these measures tend to emphasize the efficiency aspect of CT. Accordingly, there are many efforts to ground the analysis of sustainability on entropy, with a wide cross-disciplinary reach, such as in research on cities and urbanization [106]. Wessling [107] proposes a measure of sustainability that focuses on the framework of OET and considers the entropy production associated with various alternative technologies. He specifically examines direct air capture technologies in decarbonization and demonstrates that the energy inputs required to capture sufficient carbon to mitigate global warming are immense and could ultimately take the largest share of global energy production. Even if this energy comes primarily from renewable sources, the expansion of the necessary material infrastructure could have a drastic impact on biodiversity, which counts as a loss of complexity and hence leveraging entropy production. Wessling calculates an entropy account starting from the observation that DAC reduces the entropy of air mixing to zero for pure CO_2_ gas. Achieving this reduction requires substantial energy input and results in an increase in entropy production elsewhere. In out-of-equilibrium assemblages, this “elsewhere” can be found in many different locations where processes unfold to enable the technology, such as the production of solar cells for energy generation or the manufacturing of machinery used in capture and storage technology. As a general rule, any intermediary use of solar energy for processes other than direct power generation—such as in electric engines—leads to a deterioration of the entropy account. From this, we conclude that these applications are not sustainable and even accelerate entropy production in the larger assemblages.

From this discussion follows a simple and concise measure of sustainability: if growth is a necessary physical manifestation of evolution, then assemblages that coevolve can only be sustainable in the long run if they achieve a minimum rate of acceleration of entropy production. In simpler terms, slowing down evolutionary dynamics fosters sustainability, a viewpoint shared by economists [108]. In practice, this means that when comparing two technological options—such as industrial farming and organic agriculture—we should evaluate the entropy production of each within the broader context of the embedding assemblages [1]. We would then choose the option that results in less entropy production. This is generally observed in stable ecosystems, where a delicate balance is maintained between resource consumption and reproduction, which is essential for sustained production.

In the context of human economies, John Stuart Mill described a relevant concept known as the “stationary state” [109]. This state does not rule out growth or imply a decline; instead, it refers to slow growth that approaches zero growth. When discussing this vision, we should not overlook that this criterion refers to zero growth in ecological footprints rather than in monetary values. Therefore, it is possible to achieve the former without jeopardizing the latter. The reasoning is straightforward: GDP measures value added. If the material consumption of production inputs is adequately priced—meaning that the environmental costs of production are internalized—it doesn’t necessarily mean that value added cannot increase. In fact, John Stuart Mill interpreted the stationary state in a way that allowed for significant growth in spiritual and human experiences, which, when processed through markets, also translates into increased monetary value added.

As we see, PT offers a powerful framework for applying OET in the context of human economies, potentially becoming the “lived thermodynamics” of the 21st century, similar to how CT was in the 19th century. By utilizing the concept of out-of-equilibrium assemblages, we can navigate the challenges associated with directly applying thermodynamic theories to human practices. PT enables us to establish a straightforward principle for guiding design decisions and technological choices: the minimization of the rate of acceleration of entropy production, following Lineweaver’s reformulation of the Second Law.

## 6. Conclusions

The OET perspective, rephrased using assemblage theory, presents a promising foundation for developing Lived thermodynamics that encourages sustainable practices at every level. Consider, for instance, the seemingly unstoppable trend in human economies towards increased energy densities and the mass of devices, despite ongoing efforts for efficiency and miniaturization [89]. While chip manufacturers pursue miniaturization, the energy flow through these chips continue to increase, the same with storage facilities such as batteries [110]. Decarbonization is a key feature of the transition to electric vehicles; however, these vehicles often match the horsepower of racing cars, and the weight of passenger vehicles has been rising year after year (https://www.inovev.com/index.php/en/market-analyses/category-blog/19969-2023-30-2, accessed on 20 October 2025). Moreover, cities are expanding even in times of demographic decline, partly because the per capita consumption of living space has increased, resulting in larger average housing unit sizes over time. Rebound effects are prevalent in various aspects, such as in lighting [111,112].

More basically, the focus on decarbonization blinds people to the wider impact of their technological choices. There is a remarkable disjunction between the perceptions of global warming and the loss of biodiversity, which sustains the belief that if only we adapt our technology, the containment of global warming will automatically heal all other ecological crises. As we have seen, the opposite can be true: If technosphere growth is fuelled by solar energy, this does not resolve the conflict with biosphere sustainability.

A lived thermodynamics perspective on OET significantly transforms how we perceive everyday economic choices, leading to a more holistic understanding of the complexity of various assemblages. Wessling provides an insightful example regarding vegan lifestyles, in which milk and dairy products are avoided in favor of substitutes. This choice is often driven by ethical concerns, such as regarding the separation of calves from their mothers to enhance human access to milk. However, if we examine the entropy balances involved, the case against milk consumption becomes less clear-cut. The production of milk substitutes is energy-intensive and often entails agricultural practices that can be ecologically problematic. Additionally, when cows are raised according to organic farming standards and in ways that respect their species-specific needs, they play a vital role in maintaining biodiverse meadows and landscapes, beginning with the natural fertilizer of cow dung that is widely distributed by the free ranging animals. Loss of biodiversity counts as a loss of complexity and hence entropy production. At the same time, producing the milk substitutes goes along with energy dissipation and entropy production by the human technology. In other words, this form of veganism may not be sustainable in the PT view. As we see, in the Lived thermodynamics of PT the abstract notion of entropy can become meaningful in individual lifestyle choices just as the notion of efficiency in choosing energy-saving LEDs over light bulbs.

The shift from a systems view to an assemblage view has significant consequences. The systems approach implies that a system can be controlled through external interventions. In contrast, the assemblage view emphasizes that the processes within an assemblage are shaped by a complex network of interactions that vary in scope and influence, making them much less susceptible to direct control. As a result, the practical application of OET is connected to an ethical stance that promotes responsible care and caution, acknowledging the limitations of human knowledge in managing these complex assemblages [113].

## Figures and Tables

**Figure 1 entropy-27-01257-f001:**
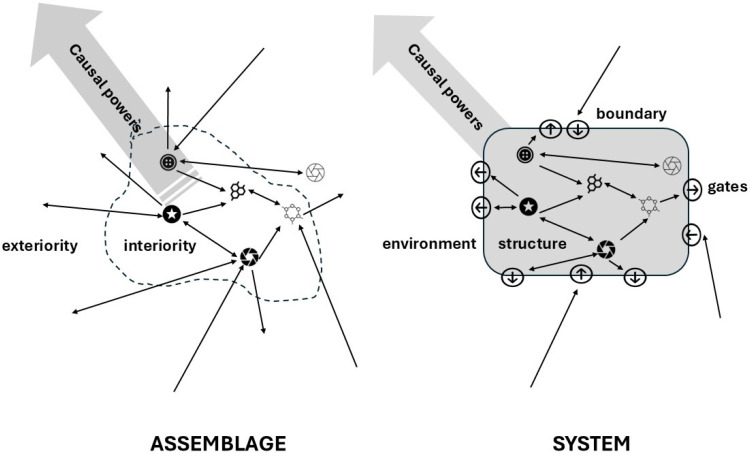
Assemblage versus system. Consider two sets of the same heterogeneous components (symbols inside the assemblage and system) that interact with each other and with various external elements (indicated by arrows). In the case of a system, these components are separated from the environment by a fixed boundary. The interactions between the components and those outside this boundary are regulated by gates whose functions are determined by the internal mechanisms of the system. These mechanisms define its internal structure, which is reflected in the interactions among components within the boundary. In contrast, an assemblage consists of the same components that maintain relationships among themselves, creating a domain of interiority without a fixed boundary. The components in an assemblage maintain autonomous interactions that extend beyond it. The assemblage manifests emergent causal powers that arise from these internal relationships (thick arrow in grey). While systems also exhibit emergent causal powers, these powers are structurally stable. Affordance theory provides several concepts to explain transitions between the two forms, such as “territorialization,” which refers to the increasing closure through boundary formation (which I do not discuss in this paper). In the scientific applications of assemblage theory (as in sustainability research [62]), the concept of “ecosystem” often is revealed as misleading. Most ecosystems are not systems but rather assemblages, such as river meadows with fluid boundaries and seasonal variations. In contrast, a mountain lake may exhibit characteristics more closely aligned with those of a system.

**Figure 2 entropy-27-01257-f002:**
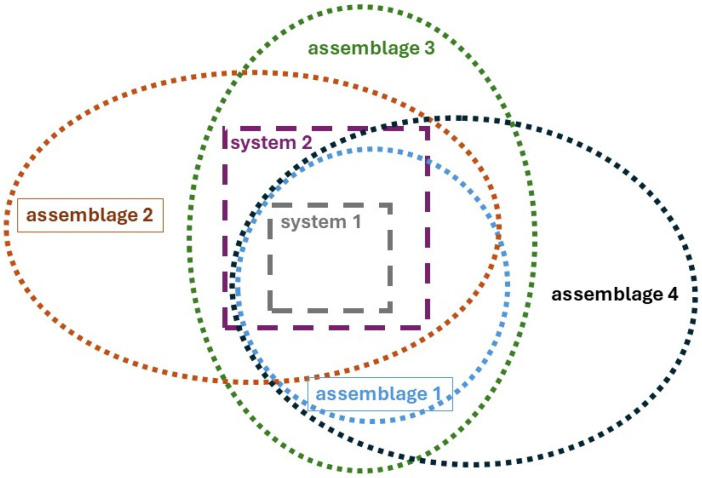
Assemblages of systems and assemblages. We continue to use both terms “system” and “assemblage.” A system is a component of an assemblage with stable and clearly identifiable boundaries. An organism or a technological artifact can be classified as a system. Systems can be hierarchically nested; for example, system 1 can be nested within system 2. The key distinction between a system and an assemblage is that a system has sub-system status, meaning it has a specific function within the larger, embedding system, which also has determinate boundaries. For instance, an engine is a subsystem of a locomotive, which is in turn part of the railway system. The human driver of the locomotive is also considered a subsystem; system 1 is the driver, and system 2 is the locomotive. At the same time, these systems are embedded within assemblages that possess a rhizomatic structure rather than a hierarchical one. The human driver is embedded in various social and natural contexts, such as a network of personal relationships (assemblage 1) or the urban ecology in which they live (assemblage 4). The driver and locomotive together are part of the railway system (assemblage 3), which itself is an element in the assemblage of regional transport networks that includes many other technologies (assemblage 4). Additionally, other important assemblages that affect the locomotive include the regional tourism network.

**Figure 3 entropy-27-01257-f003:**
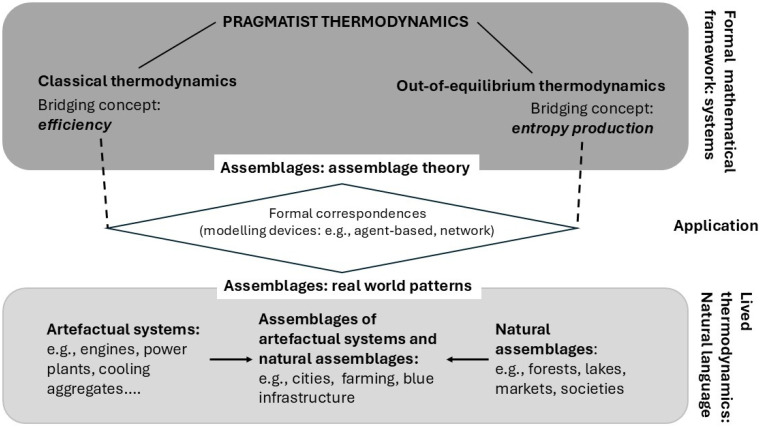
Levels of language in relating theory and lived thermodynamics. For detailed explanation, see main text.

**Figure 4 entropy-27-01257-f004:**
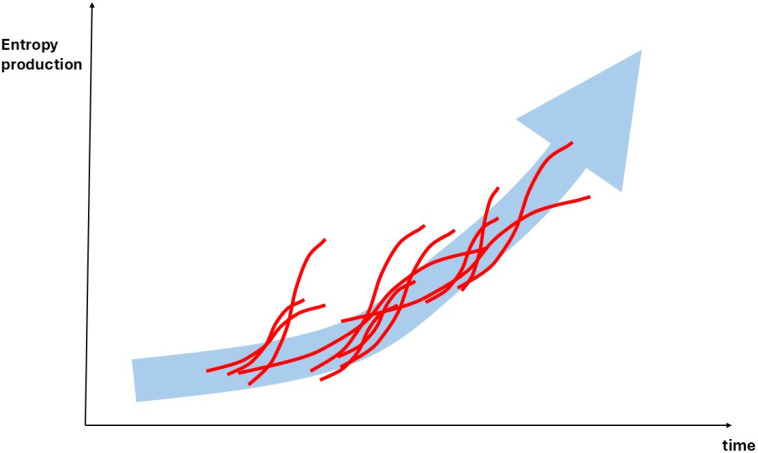
The envelope of coevolving open systems. In evolutionary assemblages, open systems coevolve in a way that their various trajectories influence one another. The qualitative diagram illustrates this fundamental pattern. Although individual systems tend to minimize entropy production, they typically show a trend of increasing entropy production during their developmental stages. In the evolutionary competition among these open systems, those in a growth phase tend to outcompete those that are closer to minimizing entropy, as indicated by sequences of events shown as crossings of the single lines of entropy production through their developmental process. As a result, we observe a dominance of systems in the growth phase within the assemblage, indicating that the larger assemblage will reflect the trajectory of these systems, ultimately leading to an acceleration in entropy production, as shown in the broad arrow. This pattern of accelerating entropy production can be vindicated by referring to the more detailed empirical literature on phylogeny, where it is driven by the so-called evolutionary transitions, such as from the prokaryotic to the eukaryotic cell, which are consistently accompanied by leveraging energy dissipation and entropy production (see below, Section 4) [41,71].

## Data Availability

No data have been used in this paper.

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
