# Peer review of "Towards Pragmatist Thermodynamics: An Essay on the Natural Philosophy of Entropy and Sustainability"

_entropy, 2025, doi:10.3390/e27121257_

Round 1

Reviewer 1 Report

Comments and Suggestions for Authors

Reviewer report

Manuscript title: Towards a Pragmatist Thermodynamics: An Essay on the Natural Philosophy of Entropy and Sustainability
Author: Carsten Herrmann-Pillath
Manuscript type: Conceptual Essay

General comments

This manuscript presents an ambitious and intellectually rich synthesis of far-from-equilibrium thermodynamics (FET), evolutionary theory, and pragmatist philosophy. The author proposes a unifying framework, so called pragmatist thermodynamics, which aims to reconnect thermodynamic reasoning with lived human experience and to illuminate contemporary sustainability challenges through an ontological and epistemological perspective.

The essay succeeds in articulating why conventional efforts on efficiency and optimization, rooted mainly in classical thermodynamics (CT), are insufficient for understanding of long-term ecological and socio-economic dynamics. By foregrounding the entropy production, evolutionary processes, and the openness of real-world systems, the author offers a conceptually persuasive account of why modern economies systematically accelerate energy throughput and entropy generation.

The manuscript is theoretically well grounded, it is clearly written, and as such also appropriate for Entropy. It provides a valuable conceptual bridge between physics, philosophy of science, and ecological economics.

In my view, it is suitable for publication after minor revisions that would further clarify several conceptual transitions and assist readers from diverse disciplinary backgrounds.

Strengths of the manuscript:

Innovative interdisciplinary synthesis. The integration of Peircean pragmatism, Lotka’s maximum power principle, and contemporary FET is original and compelling.

Clear overarching argument. The move from “systems” to “assemblages” is well motivated and offers a productive conceptual tool for rethinking thermodynamic reasoning in relation to society and technology.

Philosophical rigor. The treatment of Peirce’s concepts (tychism, habit, finality) is accurate and illuminates their relevance for thermodynamic and evolutionary reasoning.

Relevance to sustainability. The essay articulates a timely critique of efficiency-centered approaches and shows why entropy-based reasoning is essential for sustainability debates.

Engagement with contemporary issues. The discussion of artificial intelligence and renewable energy transitions reveals the applicability of the framework to real policy concerns.

Suggestions of minor revisions, which aim to strengthen the manuscript clarity, without changing its core argument.

  • The manuscript is conceptually dense; adding brief summary paragraphs at the end of major sections would aid readability.
  • A schematic figure showing the conceptual relationships between CT, FET, PT, and assemblages would be valuable for interdisciplinary readers.
  • The argument near the end concerning veganism and dairy substitutes is interesting but would benefit from a brief clarification to avoid oversimplification for readers unfamiliar with agricultural contexts.

Recommendation

The manuscript is a thoughtful and original contribution that aligns well with the thematic scope of Entropy. I recommend minor revisions. The revisions primarily concern clarification and accessibility rather than substantive changes.

Author Response

1, All substantial changes have been marked in yellow in the document. Sections 3 and 4 have been restructured to accommodate the substantial additions to section 3 and two new diagrams.

2. The paper underwent another round of language edits.

3. The reviewer’s endorsement of the general argument is highly motivating. I produced a diagram (figure 3, lines 575ff) of the structure of pragmatist thermodynamics in a combination with assemblage theory which helped me a lot to clarify my own thinking about these matters. For example, the importance of different types of language used in representing PT and making it relevant for real life was not so clear to me, and now I devote an entire subsection to this aspect. This point is also highly significant for my response to reviewer 2.

4. Figure 3 is complemented by figure 1 which adds much detail to the distinction between assemblage and system.

5. I also followed reviewer 1’s recommendation to add short summaries to the main sections of the paper.

6. The final example of veganism has been edited.

Reviewer 2 Report

Comments and Suggestions for Authors

The article proposes a philosophical reflection on the thermodynamics, with particular attention to the distinction between classical thermodynamics (CT) and far-from-equilibrium thermodynamics (FET). The author aims to show how FET can be reinterpreted as a “lived thermodynamics.” The article offers an original contribution by integrating thermodynamics, pragmatist philosophy, and evolutionary ecosystems. The idea of “lived thermodynamics” represents an innovative perspective.

The current version of the paper is ready for publication.

Author Response

Thanks for your comments.

Reviewer 3 Report

Comments and Suggestions for Authors

The manuscript presents an engaging and ambitious discussion on the importance of out-of-equilibrium thermodynamics for understanding not only physicochemical processes but also evolutionary, ecological, and socio-economic dynamics. The author clearly distinguish between classical thermodynamics and "far-from-equilibrium" thermodynamics, highlighting how the latter provides a more suitable framework to describe ensembles of open systems participating in complex processes. Furthermore, the treatment of entropy production, together with the discussion of classical principles such as minimum entropy production in dissipative structures and the maximum entropy production principle, nicely illustrates the duality and conceptual tensions that currently exist in non-equilibrium theories. The author’s interpretation and integration of these ideas into his assemblages is particularly illuminating.

I enjoyed reading the manuscript, and I believe the ideas presented have the potential to be relevant across several fields due to the foundational nature of the proposed paradigm. Nevertheless, I have several major concerns regarding the nature, scope, and intended contribution of this essay:

Major Comments

  1. The manuscript is written as a philosophical essay, which makes the narrative engaging but raises concerns about possible misinterpretations or speculative leaps. As seen in the case of Constructal Theory, strong conceptual ideas without an accompanying formalism may lack explanatory power or scientific foundation. Conversely, Lineweaver’s work attempts to introduce simple equations and schemas to better illustrate and support his arguments.

    Could the authors provide a minimal set of equations or simple models to give the proposed framework more scientific grounding and to help readers connect the conceptual discussion with formal non-equilibrium thermodynamics?

  2. The phrase “far-from-equilibrium” is widely used but notoriously difficult to quantify, and there is no consensus on its precise meaning. For an expert audience, its use may introduce ambiguity.

    I suggest replacing it with the more general and less controversial term “out-of-equilibrium” to avoid misunderstandings.

  3. Defining the “assemblies” might add more complexity to the definition of the systems in thermodynamics. While I appreciate the perspective offered, more justification is needed.

    Could the author provide a clearer rationale or a more rigorous argument for their choice of assemblages? Should assemblages be understood as a generalized space, or do they fulfill a specific thermodynamic role?

  4. The manuscript introduces the term “knowledge,” but it is unclear how such a concept could be quantified or operationalized within thermodynamics or physics. Would it correspond to an order parameter? Changes in internal degrees of freedom?

    Bringing “knowledge” into a thermodynamic framework may even evoke broader debates, such as those concerning consciousness, which may distract from the main argument.

    I am skeptical about the use of this term in its current form and encourage the author to clarify or reconsider its role in the manuscript.

Minor Comment

  1. Lines 270–279: The discussion is particularly interesting because thermodynamic systems are often studied in isolation. For example, in active matter, variations in the chemical fuel due to particle consumption are frequently neglected. When both dynamics are coupled, one finds a synergy that can explain the emergence of configurations through feedback mechanisms, where changes in structure correspond to variations in entropy production as a function of characteristic parameters (see J. D. Torrenegra-Rico’s work on the thermodynamics of Janus particles).

    Studying the full system—including the coupled dynamics of its constituents—is crucial. The system tends to increase its entropy production rate by “tuning” parameters/coefficients characterizing each species or component (see Prof. Miguel Rubi’s work on criteria for the formation of non-equilibrium structures).

    The author may find this line of research useful if he wishes to complement his philosophical perspective with theoretical results from non-equilibrium thermodynamics that directly relate system evolution to entropy production.

Overall, the manuscript is stimulating and presents a valuable perspective. However, the major comments outlined above raise substantial concerns that should be addressed to strengthen the scientific clarity and impact of the work. I would be pleased to recommend publication once these issues have been adequately considered and clarified.

Author Response

  1. All substantial changes have been marked in yellow in the document. Sections 3 and 4 have been restructured to accommodate the substantial additions to section 3 and two new diagrams.
  2. The paper underwent another round of language edits.

3. My response to Reviewer  3 is complex, and I am grateful for the important points raised. However, precisely because the comments are so stimulating, I developed some new ideas that also build on Reviewer 1’s suggestions, which I found equally helpful.

The complexity of my response arises from two reasons.

3.1. First, Reviewer 3 suggested an extremely valuable reference in the “minor comments” (which, therefore, are definitively not “minor” for me) — the work on MNET, Janus particles, and structural feedbacks. Indeed, this reference is exemplary for making sense of the assemblage concept in current thermodynamics research at the core of the discipline, rather than just at the fringes. I am grateful for this reference, so I decided to include a longer treatment of MNET when introducing the notion of assemblage, effectively serving as an internal thermodynamic model of assemblage (lines 425ff).

3.2. Second, while this step is valuable, it poses a considerable challenge to addressing the other main comment from the reviewer: the need to add a formal component to assemblage theory. As the MNET reference shows, this is possible, but it would certainly go beyond the scope of the current paper. The formalism is much more complex than simply referencing fundamental thermodynamic formulas, as done in Lineweaver’s cited paper. Therefore, I confined myself to a concise summary of the MNT approach.

4. Additionally, my response to Reviewer 3’s demand aligns with my response to Reviewer 1. This highlights the distinction between three different types of language in PT, which is an addendum to the paper and is overviewed in the new diagram and in a separate subsection (lines 493ff). In lived thermodynamics, natural language is the critical medium, and I suggest that this is also the appropriate medium for the paper that establishes its fundamental paradigm. I make this point now right at the beginning (lines 107ff). This includes the aspect that formalism that connects theoretical thermodynamics with lived thermodynamics often works via analogy, or even metaphors cast into mathematical language, as in the crucial case of Paul Samuelson’s adoption of thermodynamics in economics. I refer to this now in lines 149ff.

5. However, I have introduced a new idea. I now distinguish between the formal language of thermodynamic theory (as is also utilized in MNET) and the crucial role of simulation models in connecting theory and lived thermodynamics via the assemblage concept. Therefore, my response to the reviewer’s demand for a formal elaboration includes not only MNET but also a new argument that simulation methods provide an adequate formalization of the assemblage concept. I refer to a notable set of contributions here, specifically Joseph Vallino’s work (lines 560ff).

In sum, the new diagram figure 3 elucidates this argument about levels of language in PT.

6. I followed the advice to switch terminology to “out of equilibrium,” which is a valid point.

7. On the point of knowledge, a further elaboration goes beyond the scope of the paper. There are two occurrences of the term. First, knowledge only plays a role in the context of scientific knowledge but does not serve as an aspect of thermodynamic theory itself. This is explicit in the early reference to Peirce where I introduce knowledge in the pragmatist understanding (lines 130ff). This stems from my explicit exclusion of information theoretic strands. However, the reviewer noticed that in my discussion of agency, the meaning of knowledge as action is invoked. I have deleted this reference and refer to biological information instead (lines 763ff).

Round 2

Reviewer 3 Report

Comments and Suggestions for Authors

The author successfully addressed all major and minor comments raised in the previous round. The manuscript is now clearer, and the intrinsically philosophical perspective is more evident. The incorporation of the suggested changes—including the minor ones—has strengthened the conceptual core and improved the overall coherence of the work.

With these improvements, the manuscript is ready for publication.